# The Attitudinal Style as Pedagogical Model in Physical Education

**DOI:** 10.3390/ijerph18020374

**Published:** 2021-01-06

**Authors:** Ángel Pérez-Pueyo, David Hortigüela-Alcalá, Alejandra Hernando-Garijo, Sixto González-Víllora, Pedro Antonio Sánchez-Miguel

**Affiliations:** 1Physical and Sports Education Area, Faculty of Physical Activity and Sport Sciences, University of León, 24007 León, Spain; angel.perez.pueyo@unileon.es; 2Department of Specific Didactics, Faculty of Education, University of Burgos, 09001 Burgos, Spain; ahgarijo@ubu.es; 3Department of Physical Education, Arts Education, and Music, Faculty of Education, University of Castilla-La Mancha, 16071 Cuenca, Spain; Sixto.Gonzalez@uclm.es; 4Department of Didactics of Musical, Artistic and Corporal Expression, Teacher Training College, University of Extremadura, 10071 Cáceres, Spain; pesanchezm@unex.es

**Keywords:** Attitudinal Style, teaching models, physical education, formative assessment

## Abstract

The continuous changes in the different legislative systems have led to the application of different ways of understanding students and generating learning in them. In this sense, an area such as physical education is not alien to the continuous changes in teaching learning and its teaching has been modified from a behavioural approach to more cognitive perspectives. In this line, the Attitudinal Style concept arises with the intention of responding to this need for evolution, a global approach to teaching, as well as the generation of materials that allow the participation and learning of all types of students, greatly favoring their involvement and/or adaptation. The objective of this study is to present the Attitudinal Style as a pedagogical model within school physical education, analyzing its characteristics, elements and main purposes. A systematic review of narrative character is used, showing the origins and fundamental elements to justify the Attitudinal Style as a pedagogical model. Thus, aspects such as the generation of positive experiences in the students; the rigorous explanation of what is being learned, adaptation to the content, context and students; the work of collaboration and/or cooperation as a means to achieve a group achievement; the transfer of learning; and the application of formative evaluation, make this model of teaching applicable, relevant and necessary in the different educational levels.

## 1. Introduction

The profound educational change that was being attempted in Spain in the 1990s established a turning point in the way students were understood and learning was generated. The aim was to achieve the integral development of the students through the work of the five capacities that make up the student body: cognitive-intellectual, affective-motivational, interpersonal relations and social insertion [1,2], leaving behind the behavioural approach of the 1970s and entering into a constructivist approach. This new concept is based on the idea that capacity is the power or potentiality that the individual has at a given moment to know, to relate to another, to act in different contexts... [3] and that it conditions the concept of competence that becomes evident when this capacity is achieved [4]. 

Physical education could not be left behind and during this period, many studies showed that the level of motivation towards physical activity and attitudes towards physical education decreased considerably with age [5,6,7,8]. This situation led to the emergence of new pedagogical and didactic approaches that emerged in parallel with the teaching styles that Mosston [9] proposed in the 1960s and which have been so relevant up to the present day. Even the way of naming these approaches has evolved [10], with the appearance of approaches such as Models of Teaching [11], Curriculum Models [12,13], Instructional Models [14] and finally Pedagogical Models [15], whose characteristics are based on the indispensable relationship between learning, teaching, content and context [16]. However, this evolution is not exempt from some controversy [17,18,19]. 

Since the 1970s, different ways of dealing with teaching have emerged, new pedagogical models for both sports and physical education in general. This is how the models of Cooperative Learning [20,21,22,23,24], Sports Education [25,26,27], Teaching Game for Undestanding [28,29,30,31] or Personal and Social Responsibility [14,32,33,34,35], among others, arose, all of them are aimed at improving the students’ attitudes and towards the adherence to the practice of physical activity. However, the evolution of models no longer focuses only on what they are, but how learning, teaching, subject matter and context are applied to and through these models [19,36]; questions that will be key in this article.

In Spain, the development of Mosston’s contributions [9] was made by Delgado [37] and his structuring of teaching styles, which continues to be taught at universities, and is also common in physical education classes today. But as in the rest of the world, in Spain different pedagogical models were also beginning to be implemented, such as Cooperative Learning [38,39,40], Sports Education [41,42], the Comprehensive Model of Sports Initiation (TGfU) [43,44] or that of Personal and Social Responsibility [45,46]. However, in this eagerness to develop new pedagogical approaches, so-called emerging models emerge [10], such as Ludotécnico [47,48], Self-construction Materials [49], Health Education [50] or the Attitudinal Style [51], which is the one we deal with in this article. 

The contribution that the Attitudinal Style makes lies in three fundamental aspects: (1) The elaboration of didactic material and learning sequences that allow the whole student body to achieve motor skills and from an integral approach, without exception, from the inclusion and managing to generate group in class. (2) It has a global approach that allows it to be developed with any type of content. (3) Evolution: the integration of training assessment and competences in the learning process itself.

For all these reasons, the objective of this article is to explain, argue and justify the reasons that justify the Attitudinal Style as a pedagogical model. To this end, its origins, elements and basic principles are considered in order to establish its justification as a pedagogical model.

## 2. Systematic Review Used

For this purpose, a revision of a narrative nature has been used, in order to integrate the diversity of elements that justify the Attitudinal Style as a pedagogical model. This type of review is truly useful within the field of education, as it allows certain realities to be detailed, justified and, therefore, understood [52]. Narrative reviews help the reader to understand the essence of the content to be shown, providing evidence and justification to the text [53]. Its use in the field of research is necessary and fundamental, as it allows facts, situations and evidence to be justified from an explanatory point of view. In this case, the contribution is significant within physical education, giving rigour to such substantial elements as methodology and evaluation. 

Therefore, this is not a traditional systematic review, but an argumentative explanation about the justification of the Attitudinal Style as a pedagogical model. For this purpose, its characteristics and main elements are presented, reflecting all the didactic and scientific production up to the moment. Therefore, it makes no sense to present the aspects related to the inclusion and exclusion criteria or the reliability of the data extraction, more related to systematic reviews of an analytical and quantitative nature.

## 3. Origins of the Attitudinal Style

In Spain, once the Transition was over, the new educational legislation generated a substantial change in education, and in physical education, the transformation was even more profound. These were years in which an attempt was made to leave behind a physical education based on physical performance and mere technical execution generated by a psychomotor elite, as well as “ineptitude for” and “avoidance of” physical activities in a very high percentage of the student body [54,55]. In an attempt to leave behind teaching styles [9,37], new models were beginning to appear, although their implementation has been slow [10,56].

For these reasons, López-Pastor [57], based on Tinning’s proposal [58], proposes to frame the teaching processes in different frameworks of rationality (between those based on the students’ performance and those involving them in participation), including the Attitudinal Style within the discourse of participation and the framework of practical rationality.

The Attitudinal Style began its journey in Astorga, a small city in León (Spain) at the end of the nineties, in three schools with different economic and social contextual characteristics, an important question since this diversity is key to initiate a model [12,30,59]. This proposal arises from its intention to respond to four questions that are considered key to the involvement of students in the learning process [60]:(a)To reflect initially on fundamental questions of the teaching and learning process that provoke the teacher to question the why and for what purpose of the decisions taken. This need for reflection is highlighted by authors such as Westbury, Hopmann and Riquarts [61] when considering the importance of models that approach teaching as a reflective practice.(b)The enormous importance of the motivational aspect [62,63], to which students it is going to be applied, what their previous experiences are and how to generate them all positively so that physical activity can be carried out autonomously in the classroom and in their free time [64]; aspects that Casey et al. [16] consider to be key.(c)The questioning of the educational functionality of technical (motor) progressions when we move in educational environments that have little or nothing to do with the performance environments in which they were designed. In this sense, the need to make decisions about what is needed, how to take it and why to take it to the educational situation seems key [65].(d)The unnecessary nature of demonstration as an essential resource for generating learning and what happens when the teacher is not able to do so [54,66].

The Attitudinal Style is characterised by its global character [65], its flexibility and its capacity to adapt to the students and their contexts, as well as its search for positive individual experiences in all the students, without exclusion, in order to generate the feeling of belonging to a group.

The benefits of the incorporation of formative assessment do not end there, since it allows students to understand the purpose of learning, what the consequences are in the short term, such as the improvement of self-efficacy [67], and to begin to understand the benefits that these have in their subsequent development. However, the greatest advantage of incorporating formative assessment into the model is that it allows teachers to carry out a process of reflection that enables them to improve their teaching practice constantly; which, without doubt, benefits the students [68]. In physical education, it has been shown that mastery of the content to be taught, as well as how to teach it, produces a significant improvement in student learning [69,70,71,72]. The Attitudinal Style is a pedagogical model that directly addresses the development of the basic psychological needs and motivation of the students. Its structure is based on the perceived competence of the students, developing motor patterns of action that allow them to be autonomous in their actions. This is associated with the promotion of social relations, since tasks are proposed that encourage both individual and group achievement with others [73]. This pedagogical model seeks to generate the intrinsic motivation of the student [74], posing tasks that represent attractive challenges for the students. Thus, and under the orientation of achievement towards the task, transferability and trans-contextual analysis of the learning generated is sought [75].

In summary, Attitudinal Style made a number of key contributions to the teaching and learning of physical education, and it has been demonstrated that this style of teaching has been successful: (a) that it is possible to generate positive experiences in all the students, without exception and from inclusion; (b) that it is essential to know why and for what purpose one wants to teach, in coordination with the context and the specific content if learning is to be generated; (c) that starting from the individual achievement acquired, with collaborative work and/or cooperation it is possible to achieve group achievement involving the whole class group; (d) that the sequences of activities appropriate to the content, the context and the students allow to achieve the successful experiences that the technical progressions do not achieve in the educational contexts due to the initial heterogeneity of the group; (e) that the planning of the work focused on the transference of the learning to other contexts and collectives provokes a greater involvement of the students in their learning; and (f) that the formative evaluation is an essential element of the learning process as it involves the students through the daily use of clear, objective and appropriate instruments, provoking a collective learning. 

## 4. Elements and Phases That Constitute It

The Attitudinal Style establishes attitudes as the vertebral element of the teaching process, with the aim of achieving better learning, through greater motivation towards physical education [60]. The fundamental intention lies in generating positive experiences of self-esteem, satisfaction, autonomous thinking, socialisation, elimination of conflicts and/or supportive behaviour, starting in the area of physical education, although trying to go beyond this through its interdisciplinary proposals. The proposal shows the possibility of developing a methodological approach based on attitudes that attends equally to all the students of the same class group, from the inclusion, offering them positive experiences and managing to create the group that they should always be [60,76]. The motor is used as a means and not as an end for the achievement, working simultaneously and in balance with the rest of the abilities that develop the individual in an integral way [1,2]. The reason lies in the fact that all students have positive experiences, without exception and from inclusion [55], generating a real group that cooperates or collaborates. This inclusion allows the students’ abilities to be maximized, generating perceived competence towards the tasks demanded by the teacher [77].

The appearance of the Attitudinal Style in the spectrum of the pedagogical models tried to cover the space that goes from the dependence of the students (with regard to the teacher) in the learning process of the traditional approaches, to the essential need of autonomy and cooperation of the students (individually and in groups) of the participatory approaches to reach the integral development of the individual. 

In this sense, the Attitudinal Style: (1) allows the introduction of the students to cooperative learning by working on common aspects of it and basic pillars such as positive interdependence, promotional interaction or interpersonal relationships; although it has already been explained why it is not cooperative learning [78]. (2) In the teaching of collective sports, it introduces the students to the understanding of why and for what purpose the sport modality is carried out, starting from the heterogeneity of the participants and the involvement of the students in the learning process itself [79,80]. (3) It encourages students to take personal and social responsibility for the work related to the development of group MFs, starting from individual achievement [81]. In summary, it could be said that the Attitudinal Style facilitates the students’ transition process from the traditional approaches of teaching physical education towards other methodologies of active and participative character. 

To this end, the aim is to vary the traditional approach to teaching initiated in the concepts so that, through the procedures, the activities appear (C P A), and to begin by creating attitudes that, through some procedures, allow the intended concepts to be achieved by giving them real meaning for the students (A P C). This is done through three key elements: the Intended Body Activities (ACI), the Sequential Organisation towards Attitudes (OSA) [78] and the Final Assemblies (MF) [54,76,82]. It is very important to note that at the OSA we begin by working in pairs, trios or quartets, organised by affinity (by friendship or non-negative relationships between them) and whose heterogeneity can be focused on any of the aforementioned capacities. The reason for this is that in the area of physical education, affective-motivational and social relations are more important in achieving individual success than motor homogeneity. For this reason, homogeneity by emotional affinity, which generally carries with it a motor heterogeneity, implies that the Attitudinal Style is based on the sequence, and not the progression, of activities [54,76]. Therefore, it is not based on the fact that the heterogeneity is focused on the motor and the grouping is established by the teacher as it happens in Cooperative Learning, but that the heterogeneity from any point of view is generated by the fact that the initial small groups of 2, 3 or 4 students join others in the daily work until they form a single work group carrying out the same activity, where it cannot be excluded [82]. Moreover, the same thing happens in the organisation of the groups in the MFs, they begin by being 5 to 7 students united by affinity, and as the didactic units pass, the groups end up growing in common projects until they are the whole class.

The Attitudinal Style, in its evolution in the last decade, integrates formative evaluation as an essential part of it [60]. The reason for this is determined by the fact that pedagogical models, in general, suffer from evaluation processes that adequately complement the learning process and student involvement in it. In this sense, formative assessment refers to any assessment process whose main purpose is to improve the teaching and learning processes [83,84,85,86]. López-Pastor [57] emphasises that, in addition, he manages to get the students to learn more, correcting errors and allowing the teaching staff to learn to work better by perfecting their teaching practice [87,88]. 

This model aims to ensure that learning is complete and authentic and, as López-Pastor [57] comments, formative and shared assessment is essential in the teaching process if this is the aim; an aspect shared by Richard and Godbout [86] who defend it as an integral part of the teaching and learning process in education. Formative assessment, without yet being called a pedagogical model, but valuing its possibility [89], clearly has the same characteristics as the models as it is also based on the indispensable relationship between learning, teaching, content and context established by Rovegno [16] or Quennerstedt and Larsson [42]. 

Precisely, the incorporation of the formative assessment processes to the Attitudinal Style [60,90], associated with self-assessment and co-assessment processes [79,91] and triadic assessment through the design and development of new assessment and qualification instruments [90,91,92,93], is one of the most significant evolutions of the model, as well as its justification. Formative assessment means understanding evaluation as something more important than the grade at the end of a process. It requires establishing feedback with the students that allows them to be aware of what they are learning, and as a consequence, to favour their involvement in the tasks [94]. This produces an increase in their motivation towards learning and the acquisition of competences [95]. Furthermore, it is necessary to address it in a shared way, allowing students to decide on it through self-assessment or peer assessment [96]. Within the Attitudinal Style, the application of formative and shared assessment is fundamental, since the student is an active agent throughout the process. They have to take organizational and structural decisions, valuing their peers as essential elements in order to achieve the objectives. To this end, the teacher is a guide who favours dialogical practices in the classroom, using certain assessment instruments that favour the self-regulation of student tasks. This evaluation system is transversal to any pedagogical model, establishing a more logical and coherent transition to the final grade.

Furthermore, the evolution of the model cannot be separated from that of the interdisciplinary and interdisciplinary Attitudes working group since its creation in 2007 (www.grupoactitudes.com) and its proposal of competencies through the so-called INCOBA Project [97,98,99] which has allowed it to move from the merely interdisciplinary proposals of the Attitudinal Style to the integration of these based on a sequencing by courses through a transdisciplinary approach valid for all areas. The fundamental elements of the Attitudinal Style are presented in Table 1.

This table presents the information related to the fundamental variables that attend to the Attitude Style. These variables range from the responsibility of the students, their motivation, the criteria for the elaboration of the groups to the way in which the assessment is implicit. This table maintains coherence with the characteristics, aims and elements presented. These variables have been chosen because they clearly reflect the essential aspects to which any pedagogical model must pay attention.

The analysis of the context and the students, as well as their previous experiences, determine the regular organization and reorganization of the didactic programs and the concretion to the group class in the classroom programming.

In general, the didactic units, minimum structures of programming in Spain and that are around 8 to 12 sessions in general, are organized in three parts: (1) a first part of basic learning focused on the acquisition of positive experiences without exclusion and in collaboration and/or cooperation with peers. In this phase, which can last between 3 and 5 sessions, the ICAs, combined with the OSA, look for general positive learning experiences, in many cases maintaining the level of motor difficulty and increasing in cognitive, affective-motivational, interpersonal relations and social insertion, moving away from the concept of traditional progression and approaching that of sequence [54,76]. (2) A second phase in which, in groups that will evolve in number as the units are passed, they will prepare a MF in which the instruments of formative assessment become key so that they can evidence the individual and group learning acquired. In addition, they will have resources to carry out self-evaluation and intra-group and inter-group peer-assessment processes, which will allow them to improve both the interaction processes and the acquired learning. (3) In this last phase, the work is focused on producing evidence of the acquired learning and carrying out triadic assessment processes: self-assessment, co-assessment and hetero-assessment [60] that allow to reach consensus in the final qualification through a shared assessment process and dialogued qualification [57].

## 5. The Attitudinal Style: An Emerging Pedagogical Model

The Attitudinal Style is based on a clear and close relationship between learning, teaching, subject and context [15] and based on why and what a model is used for as established by different authors [19,36]. But in no case should it be forgotten that the trajectory, diffusion and expansion of this model are the keys to approaching the cataloguing of the name by the scientific community as a pedagogical model; in this emerging case [10] such as: (a) Possessing a clear pedagogical identity, (b) Providing the educational community with a wide repertoire of didactic and quality materials that allow for replication, and (c) Providing the scientific community with evidence of its efficacy.

### 5.1. Pedagogical Identity

The recognition of the educational or scientific community is linked to the implementation and visibility of the model in order to have a clear pedagogical identity, identifiable in a wide (or, at least, sufficient) didactic publication that allows the understanding and reproduction of knowledge and the way of generating learning proper to the area of physical education, in which the students are the protagonists and responsible for an authentic and real learning. In this sense, both the recognition of its character as a participative methodological proposal [57] and the recognition of its cooperative character, of the involvement and participation of the students and its applicability in different contexts is included in Velázquez’s research [100], the recognition as one of the most recognised teaching methods [101], the identification as an emerging pedagogical model [19,56], or as a model that hybridises with innovative strategies and formative assessment processes [102]. This pedagogical framework presented by the Attitudinal Style has a direct impact on the professional identity of physical education teachers. This allows them to be more aware, reflective and critical in relation to how the subject is taught. This is fundamental, since when the teaching of physical education is associated with solid pedagogical principles; greater learning is generated in the students [103]. This also favours social transformation through the subject, using the body as a fundamental pedagogical tool.

### 5.2. Teaching Materials 

Having a wide range of teaching material, which allows any interested teacher to replicate and put this model into practice autonomously, is essential for any interested person to be able to replicate the model. In this sense, over the last 25 years an important and extensive amount of teaching and learning material has been produced, partially collected in Tena’s bibliographical review [104] with almost a hundred exclusive references related to the Attitudinal Style [105,106,107,108,109,110,111]. In this case, all the production on formative assessment and related competences, so important in the evolution of the Attitudinal Style in the last decade, were not taken into account [60]. However, the following are some examples of practical development where the relationship between learning, teaching, subject matter and context, as well as the complementarity with formative assessment, is observed [112,113,114]. 

Undoubtedly, the search for the perception of achievement in students and the improvement of self-concept derives from the self-efficacy generated in the learning process, for which the formative assessment and the use of adequate instruments seem essential. 

In this section, some examples of the application of the Attitudinal Style to different contents are presented and where the formative evaluation favours the verification of the global approach of the model. 

One of the topics with which the class group is usually known and the implementation of this model begins is through the didactic unit related to the elaboration of the intentional games without elimination [115]. In this unit, after experimenting with games with the teacher whose characteristics are that they are as participative and dynamic as possible, and without elimination, the students in groups design and put into practice a game. At the end of the session, in a final reflection, they all comment on the aspects to be improved, propose alternatives or improvements and identify the failures or successes that the “pupil-teachers” have committed, thus managing to improve their approach. 

In physical condition, and with respect to the career work [54,76], the pupils learn to run with a constant rhythm, which they calculate by obtaining the average of the first test of 10 min. The aim is for them to learn to run as far as possible, becoming less tired and without training (only by learning to control and regulate their running rhythm and to do a proper warm-up), introducing them to a process of self-regulation [116]. After the first test they must draw up a graph with the time per lap and identify the average time run. This graph will enable the improvement produced by what they have learnt in the training unit to be compared [117,118]. As the students have two opportunities to do the second test (the one that will form part of their grade), they will have the possibility, by looking at various types of graphs and their grade assessments, of identifying the similarity with the one that would identify their grade and thus decide whether to run again to improve the grade or to stay with the one they have done in a responsible self-assessment process. The fundamental intention is associated with learning to assume the logical consequences of the work.

For collective sports such as football [80], or basketball [79], the work is very similar and divides the unit into four parts. In the first part (sessions 1, 2, 3, 4, 5), we begin with the first session by analysing and evaluating the situation of the class group in relation to football. In sessions 2, 3, 4 and 5, the class is organized in 4 groups of 6–7 students, taking into account the heterogeneity in relation to the knowledge of the sport and the technical-tactical domain, as well as the affinity between the members; and the whole process starts with decisions agreed by all, a key aspect in this model. At this point, work begins with stationary scouting (symbolism) cards and by which the groups rotate every 7–8 min, where the group self-regulation process of learning is very important [80]. In the 3rd part (sessions 10, 11), activities will be carried out where students individually and in groups will show their acquired learning and scouting knowledge [80]. In addition, in the 4th part (session 12), marking activities will be conducted [80].

The last example is the work with shadow theatre [109]. In this case, the three basic working phases of the didactic unit are still maintained. In the first one, control and mastery of the shadows and the distances to the focus are acquired in order to be able to make the figures. The fundamental characteristic is that the figures and their learning follow the strict process of the OSA. They begin by learning in pairs, so that the figure is individual for the partner to give feedback. The couple joins another couple in groups of four, so that the figure to be made is of two or three people and one or two provide the feedback. The group of four joins with another group of four to form a group of eight, and so begins with the large group figures, until the MF starts to prepare the figure by telling a story and ends with the final MF presentation for the final score, as well as its presentation in public afterwards. Other examples in different contents are: adversary sports [106], dramatisation [105], street work [104], acrobatics [110], activities in the natural environment such as rope rope work and crossing obstacles [111,119]. Evaluation and qualification tools are also provided on the Grupo Actitudes website: https://www.grupoactitudes.com/ [120].

As can be seen, there are many didactic examples approached from the Attitudinal Style. In addition, these examples deal with a diversity of content, which reflects the wide range of teaching possibilities that this model has in PE. It is a pedagogical model that focuses on the transversality of the contents: sports, physical condition, corporal expression, natural environment... emphasizing inclusion, the motivational climate of the group and the generation of positive experiences in the student as key elements. The educational experiences proposed, all of which are published with free access for teachers, show that this model provides resources to teachers on how to do things, thus encouraging much-needed reflection in the school.

### 5.3. Scientific Publication

Without a doubt, the accreditation of national and international scientific publications that demonstrate the pedagogical and didactic foundations proposed by the model is essential. In this sense, scientific research into the Attitudinal Style began at the end of the nineties and led to the author’s doctoral thesis [52] in which he proved, through the Attitude Scale for Integrated Physical Education (EAEFI), the improvement of the attitude of the students towards Physical Education. After this initial research, a decade was devoted to generating pedagogical and didactic production that would allow the model to be replicated, and five years ago it was decided to continue with the research to establish the evolution of the model, starting with the incorporation of formative assessment into the model [60]. 

Therefore, in relation to the responsibility in student assessment, Hortigüela et al. [121] start by demonstrating that the Attitudinal Style, when compared with the traditional methodological approach, significantly influences the students’ perception. Besides, Hortigüela, Pérez-Pueyo and Salicetti [122] also analysed what happened when applying the formative evaluation processes in the students who received the Attitudinal Style methodology, verifying that the perception that the students have about the evaluation received along the school year, significantly affects three fundamental factors: (a) the individual and group responsibility, (b) the work regulation during the process and (c) the authenticity of the acquired learning linked to real life. In this same research, it was verified that the students who participated in the process of distribution of grades (an essential activity in the processes of intragroup self-evaluation) [123] perceived the learning process to be more guided and coherent, although it required greater responsibility, both for oneself and for the rest. Perhaps one of the most important aspects of this issue, and one that is key to the Attitudinal Style, is that it demonstrated that the students who worked in groups most often perceived the distribution of marks as a strategy that facilitated more authentic learning. 

In this same sense, and continuing with the analysis of the importance of assessment and the relationship with the model, Hortigüela, Pérez-Pueyo and Fernández-Río [124] demonstrated how the methodological approach used by the teacher influenced the perception of physical education students with regard to their level of responsibility in the assessment process [125,126]. After the implementation of a didactic unit of acrobatics, some groups under a technical-traditional methodology and others under the Attitudinal Style, the results revealed that the perception of responsibility on the evaluation process increased significantly in the group that experienced the Attitudinal Style, which indicates the incidence of the model in relation to the use of formative evaluation processes. 

Subsequently, and continuing with the analysis of the effects of the prolonged use of a traditional teaching approach and the Attitudinal Style, it was found that students who experienced the Attitudinal Style perceived physical education class as significantly more useful and developed a stronger empathy towards the teacher than with the traditional approach. In addition, three key ideas emerged for the justification for being a pedagogical model: (a) the transcendent and fundamental teacher-student connection that it produced in the teaching-learning process, (b) the relevance of how the contents are organised, according to the needs and characteristics of the students and their sequencing in order to transmit a coherent learning process, and (c) the relevance of the transfer of learning that took place [127].

At the same time, the perception of the secondary school students, after having received a physical condition teaching unit under the Attitudinal Style methodology, was analysed in relation to the factors implicit in the physical self-concept [128]. In this unit, its effectiveness in relation to the positive influence on girls was verified. This physical self-concept is fundamental for physical education students to feel competent in carrying out motor tasks [129].

Continuing with the verification of the idea of balancing the students’ previous experiences in a clearly unbalanced content in their preference, the effects of the Attitudinal Style compared to a technical-traditional approach in the teaching of football were analysed in relation to the students’ and teachers’ perception of the classroom climate [130]. The research showed how groups of students who experienced the technical-traditional approach significantly increased the perception of an ego-oriented classroom climate. However, the groups that experienced the Attitudinal Style developed a significantly different and more task-oriented perception of classroom climate. 

In this sense, Pérez-Pueyo et al. [53] confirm that the implementation of the Attitudinal Style in future teachers meant an increase in rigour and coherence in physical education classes, highlighting the importance of groupings, the type of activities proposed by the teacher, the active role of the teacher and the individual and group responsibility of the student.

It has been observed how there is a clear connection between the didactic examples presented in the Attitudinal Style and their scientific evidence, analyzing variables that attend to the social, motivational and educational level of the student. The scientific publications of this model have the main objective of being useful to teachers in their professional performance, giving them resources that allow them to attend to the multitude of variables existing in PE from the pedagogical treatment of the body.

The research carried out in relation to Attitudinal Style has demonstrated the capacity to guarantee its replicability in different contexts, the fundamental basis of which is centred on the extensive quantity and quality of the teaching materials which support it since its appearance in the nineties. This replicability in different contexts is a fundamental requirement for any pedagogical model [131]. The Attitudinal Style has been applied for decades in a variety of educational contexts. This expansion of the model is justified in two main aspects:Initial teacher training in the Faculties of Education and of Physical Activity and Sport Sciences. The Attitudinal Style has been included as a pedagogical model within the subjects of physical education teaching pedagogy that make up the curriculum.Preparation of future physical education teachers in the system of access to the public service.

This has meant that the essence of the model has been able to be replicated in a variety of contexts. PE teachers who have received the Attitudinal Style in their initial training later replicate it in their professional practice. These professional contexts, especially at the beginning, are often very varied, and they may work in schools with very different social and economic levels. Future teachers have been trained in the Attitudinal Style for over 20 years, which guarantees the extension and dissemination of the model.

Throughout the manuscript, a variety of practical examples of the Attitudinal Style in the subject of physical education have been presented. It is a fully transversal pedagogical model, whose applicability is reflected in:-Development of any type of curricular content: physical condition, body expression, activities in the natural environment, development of motor skills...-Improved understanding of the technical and tactical elements of sports.-Involvement of students in the teaching process.-Application of self-regulation processes to generate student awareness of what has been learned.-Improvement of socialization processes in the classroom.-The inclusion of students is encouraged, guaranteeing their positive motor experiences, both individually and in groups.-Search for transferability of learning outside the classroom, especially in the generation of adherence to physical activity.-It can be hybridised with any other pedagogical model.

However, there are a number of limitations to its use:-The need to generate student autonomy towards learning.-Starting from an initial training of the teaching staff that understands the use of motor skills as a fully transversal aspect.-Need to generate more international scientific literature to show its benefits.

## 6. Conclusions

The examples of didactic development and research presented are only a few examples that show that the Attitudinal Style (and its close relationship to formative assessment) should be considered a pedagogical model. The fact that it can be applied in different contexts, adapting itself to the students, linked to any content and to the learning that is intended to be acquired, allows this statement to be made, above all, because of its usefulness for teachers to improve their educational practice.

### Final Reflection

In this sense, several important issues can be observed. These aspects reflect the importance of the Attitudinal Style as a pedagogical model transversal to a diversity of fully educational aspects. All are fully in line with the aims and fundamental elements of the Attitudinal Style analyzed throughout the manuscript.(a)The Attitudinal Style requires a high level of responsibility on the part of the student; aspects which, as they become competent in its use for the generation of learning in the formative process, they value it very positively.(b)The fact that the model allows and favors the students to self-evaluate, after generating successful experiences, makes them aware of their own learning.(c)Group work associated with co-evaluation or peer assessment processes becomes an ideal way to generate social relations and continue producing learning among peers with a cooperative or collaborative spirit.(d)These situations allow students to be more aware of the importance of motor skills to overcome fears/insecurities through successful educational actions and to achieve collective challenges, but always from the individual achievement, that generates a greater perceived self-efficacy.(e)Students are able to self-regulate their tasks and achieve quality in their productions and MF that allow them to generate the group feeling that is the basis of this model, without exclusion.

The search for the perception of achievement in the students and the improvement of the self-concept is key in the Attitudinal Style and it derives from the self-efficacy generated in the learning process. In this sense, the importance of formative evaluation and the use of appropriate instruments to achieve the completion and generation of successful experiences in physical education classes with a clear transference out of the classroom has been presented.

As future lines of research, it would be fundamental to generate more empirical evidence about the Attitudinal Style, showing its educational effects in a diversity of variables and contexts. There is still a long way to go in the world of physical education, although it should not be denied that the Attitudinal Style, as an emerging pedagogical model, is here to stay and should be considered as such.

## Figures and Tables

**Table 1 ijerph-18-00374-t001:** Fundamental elements of the Attitudinal Style.

	Attitudinal Style
Responsibility	Of a group and individual character. There is usually positive interdependence between students.
Session model	Start-up activities, intentional body activities, wedges of interest and reflections during the end of the process.
Methodological elements	Work in heterogeneous groups, even if they start out homogeneous, with a cooperative or collaborative character, through processes of intragroup self-assessment and co-assessment. Transfer to the daily and/or functional.
Time perspective	Long term.
Social skills	Maximum: communication, dialogue, active listening.
Motivation	Intrinsic and in some cases extrinsic.
Number and type of activities	Few and selected, always with the possibility of being carried out jointly by the whole group and with the help of its members.
Grouping criteria	Organization of groups that are heterogeneous in terms of technical ability and affinity in the interpersonal relations of group members, without exclusion.
Level of achievement	All students must go through all the roles to demonstrate their level of achievement and group involvement at the end. The achievement of the group is subject to the individual achievement of each member.
Goals	Shared and pre-defined.
Outcome of the process	Following individual achievement/success, there must be a collaborative or cooperative outcome.
Key capabilities	Integral development of the 5 capacities. Physical condition is the means and not the end.
Relational and emotional aspects	Starting point for the integral development of all. The characteristics of the group members condition the final presentation of the result by consensus.
Assessment	Of a formative and shared nature. It is clearly integrated into the teaching process.

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
