# Peer review of "The Attitudinal Style as Pedagogical Model in Physical Education"

_ijerph, 2021, doi:10.3390/ijerph18020374_

Round 1

Reviewer 1 Report

Dear authors,

Some efforts were done in order to improve the quality of the paper. However, unfortunately, the paper seems not to reach the high standards of the IJERPH journal. Limitations and practical implications were not clearly presented.

Author Response

Dear reviewer;

"Some efforts were done in order to improve the quality of the paper. However, unfortunately, the paper seems not to reach the high standards of the IJERPH journal. Limitations and practical implications were not clearly presented."

Reply: We have adapted to the quality standards of the journal, specifying that it is a review of a pedagogical model in Physical Education. It should be borne in mind that it is a manuscript that is based on a fully educational framework, and that contributes directly to the aims of the special issue it is aimed at.

However, at the end of section 5.3, the main limitations of the model and its practical examples have been established.

We hope that you will now consider the manuscript suitable for publication in the journal.

Reviewer 2 Report

This paper is an original and relevant education research. The article does not smoothly reveal the effects of formative assessment. Formative assessment is demonstrated in a fragmentary, though in Keyword is mentioned. 

In  Physical Education formative assessment is have novelty effect. You need described in detail.

Author Response

Dear reviewer;

"This paper is an original and relevant education research. The article does not smoothly reveal the effects of formative assessment. Formative assessment is demonstrated in a fragmentary, though in Keyword is mentioned. In Physical Education formative assessment is have novelty effect. You need described in detail."

Reply: Thank you very much for your comments.
We have paid specific attention to your considerations.
We have prepared two documents:
1- Manuscript with the changes made in red.
2- Report for the editor and reviewers in which each of the established suggestions is answered.

We hope that you will now consider the manuscript as optimal for publication in the journal.

Kind regards

Reviewer 3 Report

In this document it is presented the Attitudinal Style as Pedagogical Model in Physical Education. It is always welcoming to see a study addressing new teaching styles and their influence in students attitudes, motivation, etc. I read this paper with great interest, and my principle thought is that the paper is rather hard to read for someone that is unfamiliar with the Attitudinal Style (AS).  In addition, one of my major concerns in this paper is the use of references published in non-English journals that are also inaccessible for readers due to papers are printed. In addition, those journals are not in Scopus or Web of Science, being questionable their quality. I recommend authors to modify the document and include references of high quality jounals. Prior publication, additional issues should be addressed:

Comment 1: As a non-expert reader in the Attitudinal Style (AS), I really appreciate if authors could include a brief paragraph explaining this teaching style. As I could read in the AS, it seems that influences (or is related to) the Basic Psychological Needs, and motivational theories. If I am correct, this information should appear in the document.

Comment 2: One reference is included in line 181, please review the format.

Comment 3: Lines 210-213. It is mentioned that formative assessment is included in AS, why is it important? Please, specify that point. Up to this part, there is a lack of flow in the document being hard to non-understandable by readers.

Author Response

Dear reviewer;

Thank you very much for your comments.
We have paid specific attention to your considerations.
We have prepared two documents:
1- Manuscript with the changes made in red.
2- Report for the editor and reviewers in which each of the established suggestions is answered.

"In this document it is presented the Attitudinal Style as Pedagogical Model in Physical Education. It is always welcoming to see a study addressing new teaching styles and their influence in students attitudes, motivation, etc. I read this paper with great interest, and my principle thought is that the paper is rather hard to read for someone that is unfamiliar with the Attitudinal Style (AS). In addition, one of my major concerns in this paper is the use of references published in non-English journals that are also inaccessible for readers due to papers are printed. In addition, those journals are not in Scopus or Web of Science, being questionable their quality. I recommend authors to modify the document and include references of high quality jounals. Prior publication, additional issues should be addressed: "

Reply: Thank you very much for your words. It should be remembered that this manuscript is addressed in a special issue related to pedagogical models in Physical Education, so readers will be familiar with the subject. The Attitudinal Style originates from Spain, so some of the references come from this country. However, efforts have been made to make all of them accessible. Nevertheless, and in accordance with your suggestions, more international references have been included throughout the manuscript.

"Comment 1: As a non-expert reader in the Attitudinal Style (AS), I really appreciate if authors could include a brief paragraph explaining this teaching style. As I could read in the AS, it seems that influences (or is related to) the Basic Psychological Needs, and motivational theories. If I am correct, this information should appear in the document."

Reply:Thank you very much for the comment. This information has been included in the third section.

Comment 2: One reference is included in line 181, please review the format.

Reply:Thank you very much. We have revised the format of the reference.

Comment 3: Lines 210-213. It is mentioned that formative assessment is included in AS, why is it important? Please, specify that point. Up to this part, there is a lack of flow in the document being hard to non-understandable by readers.

Reply:This paragraph has been made more specific by giving examples of why the application of formative assessment within the Attitudinal Style is important.

We hope that you will now consider the manuscript as optimal for publication in the journal.

Kind regards

Editor's Note: we transferred your opinion about the self-citation issue to authors, authors' reply is as follows:

Authors' reply: One of the authors of the manuscript is the creator of the Attitudinal Style, which justifies the use of so many self-citations. In fact, we consider that these quotations allow us to give rigour and identity to the pedagogical model from its origins. Furthermore, we have included a variety of international references that complement them.

Round 2

Reviewer 3 Report

Dear authors,

Thank you for taking into consideration the comments.

Kind regards

This manuscript is a resubmission of an earlier submission. The following is a list of the peer review reports and author responses from that submission.

Round 1

Reviewer 1 Report

Dear Authors,

The present manuscript has merit. Overall, it is a well-written manuscript; however, some relevant issues were not taken into consideration. In this way, please find below, some suggestions in order to improve the quality of the manuscript:

- In the abstract, the goal of the study is not clear. Please review.
- Number of pages are just needed when a citation is included in the text. Please review several examples in p.6 where the number of pages are not needed to mention (e.g., “…symbolism) cards [66] (pp. 149-176)”).
- More concrete examples of didactic and research in Physical Education contexts should be presented.
- Perhaps, a graphical representation of the Attitudinal Style (including the extension of the formative assessment dimension) seen as pedagogical model is needed to better understand the interconnections between the relevant dimensions. Also from the organizational point of view, some tables could be useful to include to facilitate the work of the reader.
- There are many approaches to conduct a literature review (e.g., narrative, systematic, semi-systematic, integrative). Please clarify, from the methodological point of view, what kind/type of review/analysis was done in the present review paper and why. Detailed justification is needed. A new section perhaps could be useful to include.
- A list of (important) issues should be avoid in the “Conclusions and final reflection” section. Please review.
- The authors tried to present the Attitudinal Style as an emerging pedagogical model. Extensive work has been done by the authors and in Spain, in particular. However, it is not clear the potentiality and/or transferability of the Attitudinal Style in various sociocultural contexts. Please clarify and elaborate.
- Practical implications should be explored and presented, taking into account the scope and the high quality standards of the Environmental Research and Public Health International Journal.

Author Response

Dear Reviewer. Thank you very much for your comments. They have been of real help to us in improving the quality of the manuscript. We have uploaded two documents to the platform:
1- Reviewers' response report
2- Manuscript, in which the changes made are indicated in red.

Thank you for your review and suggestions for improvement. We believe that after modifying the manuscript, taking into account the comments indicated, the article has undergone a notable improvement. Below we explain in detail each of the changes made.

Q1: The present manuscript has merit. Overall, it is a well-written manuscript; however, some relevant issues were not taken into consideration. In this way, please find below, some suggestions in order to improve the quality of the manuscript.

Reply: Thank you very much for your words. We have responded to each of your comments in order to make the manuscript of higher quality.

Q2: In the abstract, the goal of the study is not clear. Please review.

Reply : We have reviewed the objective.

Q3: Number of pages are just needed when a citation is included in the text. Please review several examples in p.6 where the number of pages are not needed to mention (e.g., “…symbolism) cards [66] (pp. 149-176)”).

Reply: All these quotations from the manuscript have been reviewed.

Q4: More concrete examples of didactic and research in Physical Education contexts should be presented.

Reply:More concrete examples of didactic and research in Physical Education contexts should be presented.

Q5: Perhaps, a graphical representation of the Attitudinal Style (including the extension of the formative assessment dimension) seen as pedagogical model is needed to better understand the interconnections between the relevant dimensions. Also from the organizational point of view, some tables could be useful to include to facilitate the work of the reader

Reply:We have included a table reflecting the key elements of the Attitudinal Style, including the application of formative and shared assessment.

Q6: There are many approaches to conduct a literature review (e.g., narrative, systematic, semi-systematic, integrative). Please clarify, from the methodological point of view, what kind/type of review/analysis was done in the present review paper and why. Detailed justification is needed. A new section perhaps could be useful to include.

Reply: A section has been included to justify the systematic review used.

Q7: A list of (important) issues should be avoid in the “Conclusions and final reflection” section. Please review.

Reply: We are aware that the conclusions of a manuscript are not usually presented in this way. However, as this is a narrative review, we have considered doing it this way in order to make clear to the reader the justification of the Attitudinal Style as a pedagogical model.

We have tried to justify this in the conclusions section.

Q8: The authors tried to present the Attitudinal Style as an emerging pedagogical model. Extensive work has been done by the authors and in Spain, in particular. However, it is not clear the potentiality and/or transferability of the Attitudinal Style in various sociocultural contexts. Please clarify and elaborate.

Reply: An attempt has been made to justify this transferability at the end of section 5.3.

Q9: Practical implications should be explored and presented, taking into account the scope and the high quality standards of the Environmental Research and Public Health International Journal

Reply: Practical implications have been included in the new references included, both didactic and research.

Kind regards

Reviewer 2 Report

This manuscript is solid and well-written.

The authors provided a strong theoretical background to support the framework of the Attitudinal Style. 

In section 4, The Attitudinal Style: An emerging pedagogical model, two subsections were presented 4.1 and 4.2. The authors provided a very detailed review of 4.2 Teaching material, but a very limited review of 4.1 Pedagogical identify.

The authors might want to provide a balanced amount of information regarding these two sections. For example, there were many examples that were provided in 4.2, but barely in 4.1.

In the section of conclusion and final reflection, the authors might want to provide concrete recommendations for future studies regarding validating the model empirically.

Author Response

Dear Reviewer. Thank you very much for your comments. They have been of real help to us in improving the quality of the manuscript. We have uploaded two documents to the platform:

1- Reviewers' response report
2- Manuscript, in which the changes made are indicated in red

Thank you for your review and suggestions for improvement. We believe that after modifying the manuscript, taking into account the comments indicated, the article has undergone a notable improvement. Below we explain in detail each of the changes made.

Q1: This manuscript is solid and well-written.

Reply: Thank you very much for your comment.

Q2: In section 4, The Attitudinal Style: An emerging pedagogical model, two subsections were presented 4.1 and 4.2. The authors provided a very detailed review of 4.2 Teaching material, but a very limited review of 4.1 Pedagogical identify.

Reply: More information has been included in that section.

Q3: The authors might want to provide a balanced amount of information regarding these two sections. For example, there were many examples that were provided in 4.2, but barely in 4.1.

Reply: More information has been included in that section.

Q4: In the section of conclusion and final reflection, the authors might want to provide concrete recommendations for future studies regarding validating the model empirically.

Reply: These recommendations have been included.

Kind regards

Reviewer 3 Report

Title

The title reflects well the objective of the study.

Abstract

The summary is very general and does not indicate at any time the methodology used. Does not give the reader relevant information about the content of the article.

Keywords

It is recommended to use synonyms of the title words in the keywords

General comments

The bibliography used is not up to date. This leads me to wonder if the authors have actually done a literature review for years. It is a very current issue given the methodological change that has emerged in all areas of knowledge, therefore there is a lot of updated literature.

I do not know to what extent it is appropriate to talk about a certain country when you are following a general idea with an international vision.

Another perceived problem with references is the large number of citations of documents in Spanish. It is repeated again, it is not a work focused on the Spanish context so it should have focused on a more international vision.

This theoretical study would have been more complete if it had been defined as a systematic review based on the characteristics of this methodology. Thus, its objectives would have been better defined.

It gives the feeling of being a work without methodological structure that collects random bibliography without a clear meaning.

A high level of self-citation is shown.

Author Response

Dear Reviewer. Thank you very much for your comments. They have been of real help to us in improving the quality of the manuscript. We have uploaded two documents to the platform:

1- Reviewers' response report
2- Manuscript, in which the changes made are indicated in red

Thank you for your review and suggestions for improvement. We believe that after modifying the manuscript, taking into account the comments indicated, the article has undergone a notable improvement. Below we explain in detail each of the changes made.

Q1: Title

The title reflects well the objective of the study.

Reply: Thank you very much for your comment.

Q2:

Abstract

The summary is very general and does not indicate at any time the methodology used. Does not give the reader relevant information about the content of the article.

Reply: The abstract has been completed.

Q3: Keywords

It is recommended to use synonyms of the title words in the keywords

Reply: We have changed the term of pedagogical models.

Q4: The bibliography used is not up to date. This leads me to wonder if the authors have actually done a literature review for years. It is a very current issue given the methodological change that has emerged in all areas of knowledge, therefore there is a lot of updated literature.

Reply: More current bibliographical references have been included.

Q5: I do not know to what extent it is appropriate to talk about a certain country when you are following a general idea with an international vision.

Reply: As it is a pedagogical model born in Spain, its origins have been justified in the manuscript, in order to make clear to the reader its origin. However, current references of an international nature have been included.

Q6: Another perceived problem with references is the large number of citations of documents in Spanish. It is repeated again, it is not a work focused on the Spanish context so it should have focused on a more international vision.

Reply: More international references have been included.

Q7: This theoretical study would have been more complete if it had been defined as a systematic review based on the characteristics of this methodology. Thus, its objectives would have been better defined.

Reply: A new section has been added, after the introduction, in which a systematic review of a narrative nature is justified.

Q8: A high level of self-citation is shown.

Reply: This is due to the justification of the origin and elements of the pedagogical model. However, more relevant references from international authors have been included.

Kind regards

Round 2

Reviewer 1 Report

The authors tried to follow the proposed suggestions/comments. Some additions were made; however, the included text does not give me sufficient confidence to proceed with positive feedback. Major issues to consider:

- Taking into account the high standards of IJERPH, scientific writing style is missing, for instance: “public, private, rural and city schools... by Physical Education teachers in the course of their professional work.”

- Examples of didactic and research in Physical Education contexts were not well presented.

- The content of Table 1 is confusing.

- Still, it is not clear the potentiality and/or transferability of the Attitudinal Style in various sociocultural contexts. A clear Glocal perspective is missing; just the local is given.

- Section 2 needs support from literature review.

- Practical implications were not made correctly as expected.

Reviewer 3 Report

The authors' response is greatly appreciated. The manuscript has improved its quality considerably although we still find a methodology not clearly established. As it is a systematic review, sections such as Search strategy, Inclusion and exclusion criteria, Reliability and data extraction, Quality assessment and level of evidence and prodedure must be addressed. Therefore, in my opinion the manuscript is not complete for publication.